# Cantilever Beam with a Single Fiber Bragg Grating to Measure Temperature and Transversal Force Simultaneously

**DOI:** 10.3390/s21062002

**Published:** 2021-03-12

**Authors:** Abdulfatah A. G. Abushagur, Norhana Arsad, Ahmad Ashrif A. Bakar

**Affiliations:** 1Center of Advanced Electrical and Communication Engineering, Faculty of Engineering and Built Environmental, Universiti Kebangsaan Malaysia, UKM Bangi, Selangor 43600, Malaysia; abushagur@ukm.edu.my (A.A.G.A.); ashrif@ukm.edu.my (A.A.A.B.); 2Department of Electrical and Electronic Engineering, Faculty of Engineering, University of Gharyan, Gharyan, Libya

**Keywords:** fiber Bragg grating, strain, temperature, discrimination, cantilever

## Abstract

This work investigates a new interrogation method of a fiber Bragg grating (FBG) sensor based on longer and shorter wavelengths to distinguish between transversal forces and temperature variations. Calibration experiments were carried out to examine the sensor’s repeatability in response to the transversal forces and temperature changes. An automated calibration system was developed for the sensor’s characterization, calibration, and repeatability testing. Experimental results showed that the FBG sensor can provide sensor repeatability of 13.21 pm and 17.015 pm for longer and shorter wavelengths, respectively. The obtained calibration coefficients expressed in the linear model using the matrix enabled the sensor to provide accurate predictions for both measurements. Analysis of the calibration and experiment results implied improvements for future work. Overall, the new interrogation method demonstrated the potential to employ the FBG sensing technique where discrimination between two/three measurands is needed.

## 1. Introduction

Fiber Bragg gratings (FBG) are emerging in a wide range of fields, especially in applications where the general strain gauges and encoders are not applicable due to the environment, wiring complexity, and electromagnetic interference [1,2,3,4]. The direct response of its resonance Bragg wavelength (RBW) to external physical effects is another critical feature that makes the FBG-based sensor a key photonic element compared with other fiber sensors. However, the RBW has its limitations through its cross-sensitivity when fiber-containing FBG is exposed to more than one external physical effect simultaneously, such as temperature with strain, axial with transversal forces, axial/transversal force together with temperature, and others.

An early solution was simply via adding reference FBG to the primary sensing one, for which the second measurand should be compensated/measured. Another solution is to make a hybrid of the FBG sensor with different fiber grating techniques such as long-period grating [5]. However, each is no longer preferred as they add to the complexity of data acquisition and make the sensor size bigger. Many researchers have devised ways to address the issue with the aid of employing either a single specific type of FBG technique, e.g., tilted FBG (TiFBG) [6], or with the assistance of inscribing the gratings into a distinctive fiber type, e.g., Hi-birefringent fiber (PM) [7,8]. Another group manipulated the packaging mechanisms of the FBG [9,10] or simply manipulated its spectrum [11,12,13] to utilize two spectrum parameters that respond differently to, e.g., stress or/and temperature.

While these methods are appropriate for stresses applied along the longitudinal direction of the optical fiber axis, they frequently lead to complexity, are unreproducible, or packaged in a procedure that does not scale to mass-production and is cost-ineffective for huge deployment. The more interesting method that seems reproducible is based on half bonding chirp FBG to measure temperature and pressure [14] simultaneously and temperature-insensitive hydrostatic pressure [15]. Both are based on BW modulation in which the half-bonded grating will shift with strain and temperature, while the other half, which is set free, will shift only with temperature changes. However, the thermal coefficients of the half-bonded FBG will not be the same as the free ones unless the glue is carefully considered. Moreover, it is susceptible to the Fabry–Perot effect and signals distortion if not packaged well. 

However, the fiber applications where they are susceptible to bend under transversal pressure/forces have been overlooked and paid little to no attention.

In several applications and, in particular clinical disciplines, the instruments they use are subjected to bending during surgery, such as catheterization [16,17] and retinal microsurgery [18]. Researchers utilized the bandwidth (BW) method for either temperature-independent force measurement or force components discrimination, respectively. Bandwidth is the difference between the high wavelength resonance region and low wavelength resonance region, which are indicated in this work as longer wavelength (LW) and shorter wavelength (SW), respectively. Both LW and SW respond differently to transversal forces (localized strain), causing a BW change.

Instead of the conventional method utilizing the BW and center wavelength of the FBG, we investigate the new interrogation method in which the bandwidth components LW and SW are used. Using these two wavelengths, three attributes of the FBG’s spectrum would become available to measure three different measurands, e.g., tapered FBG. The tapered FBG has an inhomogeneous cross-sectional area, hence the employment of both LW and SW together with the center wavelength can be of great benefit in discriminating between two forces and temperature. They would respond differently to axial and transversal forces.

We believe that this is the first work to discriminate between transversal forces and temperature, employing the new method based on the longer and shorter wavelengths of a single FBG.

## 2. Theory and Sensing Principle

The proposed force and temperature sensing module are based on a metal ruler (indicated as a simple beam) integrated with single FBG. The beam is modeled as a cantilever beam with fixed-free boundary conditions. The beam with the FBG sensor is subjected to both transversal forces and temperature variations.

The transversal forces are applied at the beam’s tip (free end), which causes the beam to bend. According to Bernoulli’s beam theory, bending moment (Mx) to resist the bending force would be induced, as shown in Figure 1, where the beam is likely to bend along the negative y-axis.

Because the FBG has adhered to the beam’s surface (host), the strain-induced onto the beam’s surface is supposed to be transferred to the FBG. The inset in Figure 1 illustrates the cross-sectional area A as it is viewed according to the two orange arrows. Since the FBG is bonded to the beam and given its small size, it becomes a part of it that follows the host strain.

The strain along the beam’s surface is linearly dependent on the bending moment, and, therefore, it is localized and proportional to the transversal forces applied at the beam tip. The local strain can be calculated according to the beam theory as follows,
(1)$$\varepsilon \left(z\right)\;=\;\frac{M_{x}\left(z\right)y_{t}}{E_{b}I_{b}}\;=\;\frac{F_{y}zy_{t}}{E_{b}I_{b}}$$
where $E_{b}$ is Young’s modulus of the beam, $I_{b}$ is the second moment of the beam, which can be defined as $I_{b}\;=\;\frac{bh^{3}}{12}$, $y_{t}$ is the distance from the neutral axis to the beam’s surface, and *z* is the distance between the tooltip and the local segment of the FBG. The fiber elastic-optic, thermo-optic properties, and the nature of the strain-induced to the beam that the fiber is embedded within, govern the sensitivity of the FBG.

In the case of homogeneous and isotropic strain, the whole spectrum of the FBG would shift equally, by monitoring, only the center wavelength can calculate the strain-induced if the temperature is kept unchanged. However, when the strain induced is non-uniform, and it is a linear function of FBG length as in our case here, the spectrum of the FBG would respond with different sensitivity causing a local shift of the wavelengths lead eventually to bandwidth tuning.

In contrast to the induced mechanical strain, a gradient change in temperature along the short segment of the FBG sensor is unlikely to exist. Hence, the whole spectrum of the FBG responds with the same sensitivity to the temperature variations.

According to the above discussion, various spectrum parameters can provide independent information about the induced non-uniform strain and the temperature changes. Here in this work, the proposed method is to monitor the longer and shorter wavelengths as shown in Figure 2 relative to the resonance Bragg center wavelength ($\lambda_{LW}\;\&\;\lambda_{SW}$), respectively.

The amount of shift in both wavelengths due to strain and temperature variations is given by.
(2)$$∆\lambda_{LW}\;=\;k_{\varepsilon lw}{ℰ}_{LW}\;+\;k_{T}∆T$$
and
(3)$$∆\lambda_{SW}\;=\;k_{\varepsilon sw}{ℰ}_{SW}\;+\;k_{T}∆T$$
where ${ℰ}_{LW}$, and ${ℰ}_{SW}$ denote local strain at longer and shorter wavelengths, respectively, ∆T denotes the temperature change, $k_{\varepsilon lw,\varepsilon sw}$ and $k_{T}$ denote the constant sensitivity ratio associated with strain and temperature, respectively. Substituting (1) in (2) and (3) we obtain,
(4)$$∆\lambda_{LW}\;=\;k_{\varepsilon lw}\frac{F_{y}Z_{LW}y_{t}}{E_{b}I_{b}}\;+\;k_{T}∆T$$
and
(5)$$∆\lambda_{SW}\;=\;k_{\varepsilon sw}\frac{F_{y}Z_{SW}y_{t}}{E_{b}I_{b}}\;+\;k_{T}∆T$$
where $Z_{SW}$ and $Z_{LW}$ are the distances from the free end of the beam to the proximal and distal position edges of FBG sensor. Equations (4) and (5) can be expressed as follows:(6)$$∆\lambda_{LW}\;=\;k_{FLW}F_{y}\;+\;k_{T}∆T\;∆\lambda_{SW}\;=\;k_{FSW}F_{y}\;+\;k_{T}∆T$$

Here $k_{FLW}$ and $k_{FSW}$ are the proportionality constant representing the sensitivity ratio of LW and SW with respect to applied transversal forces, respectively.

Then we can use matrix notation to map the readings of the two wavelengths independently for the force and temperature calculations as follow:(7)$${\left[F_{y}\;\Delta T\right]}^{T}\;=\;K^{-1}{\left[\Delta \lambda_{LW}\;\Delta \lambda_{SW}\right]}^{T}$$
where $K^{-1}$ is a 2 × 2 inverse matrix of the transfer function matrix.

## 3. Simulation and Experimental Work

Before conducting experiments, a simulation environment was developed to investigate and predict how the beam reacts to real-world forces and temperature. The strain and force relationship along the beam is estimated to decide where the FBG should be placed, and then the time taken for the heat to transfer and reach the target temperature at the presumed FBG position. The position of FBG, therefore, can be decided accordingly based on the simulation results.

A series of experiments are then conducted to calibrate the single-axis sensor model by subjecting the beam to static transversal forces and temperature variations.

### 3.1. Simulation Work Using COMSOL Environment

The simulation is executed via finite element analysis (FEA) in COMSOL Multiphysics, where strain distribution caused by transversal forces can be accurately estimated. The corresponding three-dimensional model was built and subjected to static transversal forces ranging from (0.2–1.4 N), and temperature variation went from (25–60 °C). The main parameters which have been used in the simulation are listed in Table 1.

The distribution function of the induced strain along the beam, when subjected to 1.4 N force, is illustrated in Figure 3a. It can be seen that the maximum strain is occurring at the fixed end of the beam. Figure 3c shows the strain distribution along the beam when subjected to several transversal forces. The strain-induced due to each force magnitude gradually increases from zero at the free end to reach its maximum value, while approaching the other fixed end of the beam/ruler. The vertical dashed red lines indicate the presumed position of both ends of the FBG sensor. It can be concluded that the gratings’ elongation, therefore, would be localized, due to which chirp would be induced under bending load. It was evident that the rate change of the gratings’ elongation, which proximal to the fixed end of the beam, develops with the most significant sensitivity compared with the rate change of the gratings’ elongation in anywhere else. In other words, the shifts of the two wavelengths, LW and SW, would be of interest as both have different responsivity ratio to the strain differential. Temperature distribution and heat transfer results from the simulation are shown in Figure 3b–d. The heat is transferred to the FBG by drawing a heat sink near the metal ruler in the FBG vicinity. Based on simulation results, uniform distribution of temperature along the presumed FBG position can be achieved after some time, as shown in Figure 3d.

### 3.2. Experimental Work

The integrated beam with the FBG sensor was calibrated with an automated calibration system using the LABVIEW program. The simplified schematic version of the experimental setup of the calibration system used in this work is shown in Figure 4a. A superluminescent light-emitting diode (LED) with an optical circulator built-in is used as a broadband light source (Dense Light Semiconductors), centered at a wavelength of 1550 nm, together with an FBG interrogator of type I-MON USB (Ibsen Photonics, Ryttermarken 17, Denmark). A computer is utilized to control both through the USB interfaces. The I-MON 256 comes with a high-speed USB interface for real-time measurement up to 5 kHz. Figure 4b shows the hardware’s used in the automated system, which consists of the tri-axis motorized stages (x, y, and z), force gauge from (SAUTER GMBH) of type FH10 with a resolution of 5 mN. The temperature calibration procedure is conducted using a thermocouple (TC), temperature controller, heat element, and a heat sink, as shown in Figure 4c. The thermocouple signal is acquired by using a DAQ module from National Instruments (NI-USB-DAQ 6122).

The LABVIEW program is developed with two algorithms that enabled a more precise analysis of the data in which signal processing is performed to calculate the two wavelengths LW and SW. It is also worth mentioning that the FBG sensor was sampled in this work at a scan rate of 1 kHz due to the size of the program.

#### 3.2.1. Repeatability

The purpose of the repeatability procedure is to study the consistency of the two wavelengths’ response to the applied forces. Four calibration procedures are performed with regard to transversal forces. By contrast, for the temperature calibration process, the sensor is subjected to heating procedures only once. The measurement time of our heating process takes around 5 min (see Figure 3d) to allow the target temperature to become uniformly distributed along the portion of the beam at which the FBG segment is bonded. However, obtaining the same responsivity ratio for both LW and SW wavelengths to temperature changes would be convincing.

#### 3.2.2. Calibration Process

The FBG sensor with a length of 10 mm is bonded onto the beam’s surface (substrate) utilizing an epoxy adhesive. The sensor’s spectrum parameters, such as longer and shorter wavelengths, are monitored while exposing the sensor to the applied forces and temperature changes. The two wavelengths LW and SW are measured corresponding to the points at which the power has dropped to half of its peak value, as shown in Figure 2.

(a)Transversal forces calibration proceduresThe beam was fixed at one end while its other end subjected to the loads transversely. As shown in Figure 4b, the forces applied at the free end of the beam were controlled by the automated system. The system holds the force gauge position through the vertical translation motor (*z*-motor stage) with micron-level precision.During the calibration procedures, the force gauge is used to push the beam downward at the beam’s tip, increasing the applied force (bending the beam) from 0 N to ≈4.6 N with an interval of 1 mm translation. This procedure is then repeated four times at a constant temperature of ≈25 °C. Four subsets of data samples are collected and logged into a file for further analysis.(b)Temperature calibration procedureRegarding the calibration procedures to evaluate the sensor’s response to the temperature changes, the heat-sink to which the heating element is attached was brought underneath the FBG position touching the beam’s bottom side (see Figure 4c). The temperature at the beam proximal to the FBG sensor (measured with the thermocouple) has increased from 25 °C to 60 °C with an interval of 5 °C manually via the temperature controller.

#### 3.2.3. Calibration Results

Figure 5a shows the response of the reflected spectrums of the FBG sensor as the temperature goes up with an interval of 5 °C across the range 25–60 °C. Both wavelengths LW and SW are supposed to shift equally as the environmental temperature typically has uniform distribution, especially over a short segment. To evaluate the sensor’s temperature’s impact and sensitivity, two graphs are plotted to demonstrate the reaction of both wavelengths LW and SW as a temperature function. Figure 5b,c show the linear fitting results of both wavelengths’ LW’s and SW’s response to the temperature changes across the range of (25–60 °C), respectively.

Two insets are added to Figure 5b,c to illustrate the standard residual error of both wavelengths’ responses to the temperature changes. The most deviated observed shift value from its corresponding fitted value occurred at a temperature of 40 °C in LW. In comparison, the SW shift’s response happens at three different temperature values of 25 °C, 35 °C and 50 °C.

It is seen that the linear fitting obtained for the calibration of both are almost the same, where their responsivity ratios are found at about 10.9 pm and 10.5 pm, respectively. This comes close to the standard sensitivity at ≈1550 nm for almost every bare single mode fibre (SMF-28) containing FBG in general.

On the other hand, reflected spectrums of the FBG sensor at specific forces are illustrated in Figure 6. As beam theory suggests, bandwidth broadening is observed because the strain distribution is not uniform along the beam’s surface. The top surface of the beam to which the FBG is attached should experience a local level of stretching. Therefore, one edge of the FBG, which is proximal to the beam’s fixed end, should expand greater than the other distant edge. Figure 7a,b illustrate the responses of both wavelengths LW and SW to the applied transversal forces, respectively. The dashed line shows their average linear fitting results.

The following matrix represents the sensitivities’ coefficients, which are obtained after averaging the results of the calibration curves for both temperature and transversal forces:(8)$$k\;=\;\left[\begin{array}{cc} 0.0712 & 0.0107\\ 0.0573 & 0.0107 \end{array}\right]$$

Both LW and SW respond to the transversal forces with different sensitivities, although temperature sensitivity is the same. The I-MON interrogator has wavelength resolution around ≥1 pm, which means force resolution of better than 17 mN and around 0.1 °C can be detected by utilizing the two LW and SW wavelengths. These coefficients are then keyed in the system. As stated in Section II, the linear mapping coefficients can be determined for the transversal force and temperature calculation algorithms. However, although both wavelengths’ shifts exhibit good linearity and strong correlation, the SW showed virtually no response for forces applied between 2 N and 3 N, which occurred consistently in the four calibration experiments.

It was noted that both wavelengths’ response exhibits less sensitivity than can be inferred from the induced strain produced by the simulation work. The drawbacks are attributed to the gluing process of the sensor. It was also noted that it remained soft even after being cured. The explanation might be that although the proportion of the hardener to the resin epoxy during the mixing phase ought to be comparable, rather, the proportion of both could have been generated in an unequal proportion due to human error, which consequently can make the FBG loose and not precisely obey the strain induced by the host substrate.

As a measure of repeatability, for each wavelength, the residual error of its shift in each experiment are calculated and then combined with the other three results to calculate the mean and standard deviation. Figure 8a,b illustrate the probability distribution of the wavelength shifts residual errors of wavelengths LW, and SW, respectively. The standard deviations were found to be 13.21 pm and 17.015 pm for the LW and SW, respectively.

The I-MON256 provides wavelength repeatability of 5 pm at maximum, and the range of the forces applied in our experiments span to approximately 5 N. Therefore, the achieved standard deviations of the proposed method were considered adequate, and due to relatively large error exhibited by the SW’s response, moderate reliable repeatability could be concluded.

#### 3.2.4. Validation Experiments

The purpose of this experiment was to validate the results obtained from the calibration experiments. Based on Equations (6)–(8), the simultaneous measurement of both force and temperature can be calculated as follows:(9)$$\left[\begin{array}{c} F_{y}\\ ∆T \end{array}\right]\;=\;\left[\begin{array}{cc} 71.942 & -71.942\\ -385.262 & 478.72 \end{array}\right]\left[\begin{array}{c} ∆\lambda_{LW}\\ ∆\lambda_{SW} \end{array}\right]$$

In this experiment, due to our available temperature control system and the time required for the temperature to distribute evenly along the FBG sensor, it was found that varying both temperature and forces simultaneously is not practical. Thus, two procedures were performed to demonstrate the discrimination between the two measurands, transversal forces and temperature. In each process, one measurand was changed while the other one was fixed at the predefined value, and the effect was then observed.

In the first procedure, the temperature was kept at room temperature while forces were applied. With the heat sink apart, a thermocouple placed close to the beam without touching it and then the force gauge was moved downward at the beam’s tip, causing the latter to bend in steps. The sensor readings of both temperature and force, as well as the force exerted on the force gauge due to the beam’s stiffness and temperature from the thermocouple were recorded in each step. An ideal sensor would measure the room temperature accurately and should retain the same reading while the detection and reading of the different forces are applied. In the second experiment, the force was applied first and then the heat sink was placed underneath close to the beam without touching it. Temperature was then varied through the temperature controller. Once more, an ideal simultaneous measurement would suggest that the sensor detects and measures both the temperature values’ changes and retain the same force value applied initially. Figure 9a shows the forces applied versus temperature variations in both experiments. The vertical data represent the first experiment, while the horizontal ones represent the second. As stated in the legend, the red marks depict the sensor readings for both experiments, and the dashed lines describe the actual values of temperature and force, respectively. The estimated force and temperature by our sensor were computed based on the LW and SW shifts and the matrix $k^{-1}$ (9).

The vertical data are comparing our sensor reading of temperature with the actual values acquired from the thermocouple while measuring the forces applied. It can be seen that the sensor can provide the changes in forces from 0–≈2 N simultaneously while measuring temperature close to the one provided by the thermocouple. On the other hand, the horizontal data confirm that the sensor can detect and maintain the same reading of the force as close as those measured by the force gauge simultaneously with the temperature changes from ≈24 °C up to ≈42 °C. Intentionally, actual values of forces and temperatures in the first and second experiments are excluded in Figure 9a for graph clarity.

Figure 9b,c comparing the sensor readings with the actual values of forces applied and temperature changes, respectively. An ideal fit would generate a straight line through the origin with a slope of 1. The sensor readings in both experiments are consistent with the actual reading except for some outliers’ values in the temperature graph.

As discussed in the previous section, the sensor beyond 2 N is not functioning well due to the glue, and inaccurate results are observed. To solve this issue, epoxy resin mixing nozzles suitable for two-component epoxy resin with proper proportion should be used in future work. Furthermore, the appropriate environmental temperature will be considered in future work to include the sensor’s dynamic response for various temperature and forces values simultaneously.

It is worth mentioning that the proposed method can be applied in any applications where the strain induced along the FBG structure is non-uniform. However, for the uniform strain, this method is not applicable.

## 4. Conclusions

We have proposed and demonstrated a new interrogation method for the first time. The longer and shorter wavelengths of a single FBG were utilized for discrimination between temperature and applied transversal forces. A single FBG was attached to a flexible beam to create a simple FBG sensor module. An FEA was carried out in simulation to investigate the force–strain relationship, strain distribution along the beam, and heat transfer rate. By that means, we could predict the proper position of the FBG’s sensor, and the time required for the temperature changes to distribute evenly with the FBG sensor. A series of experiments were performed with the sensor module. The longer and shorter wavelengths exhibited different sensitivities during transversal forces with values of 0.0712 nm/N and 0.0573 nm/N, respectively. Meanwhile, sensitivity was the same for both in response to the changes in temperature with responsivity ratio of 0.0107 nm/°C. A force and temperature calculating algorithm utilizing the sensitivities coefficients in a matrix form can estimate the transversal forces and temperature with adequate accuracy.

The investigation of this proposed method is carried out here to provide other ways of utilizing different FBG parameters to avoid the latter’s intrinsic cross-sensitivity. Furthermore, it may be possible to use the same approach for measuring axial, transversal forces, and temperature, which will be explored in future work.

## Figures and Tables

**Figure 1 sensors-21-02002-f001:**
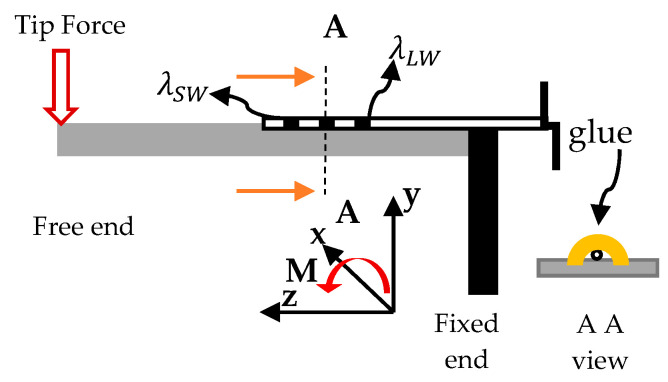
A schematic shows the beam (substrate and fiber including fiber Bragg grating (FBG) and bending moment according to beam theory, inset AA view showing the sensor buried underneath the glue.

**Figure 2 sensors-21-02002-f002:**
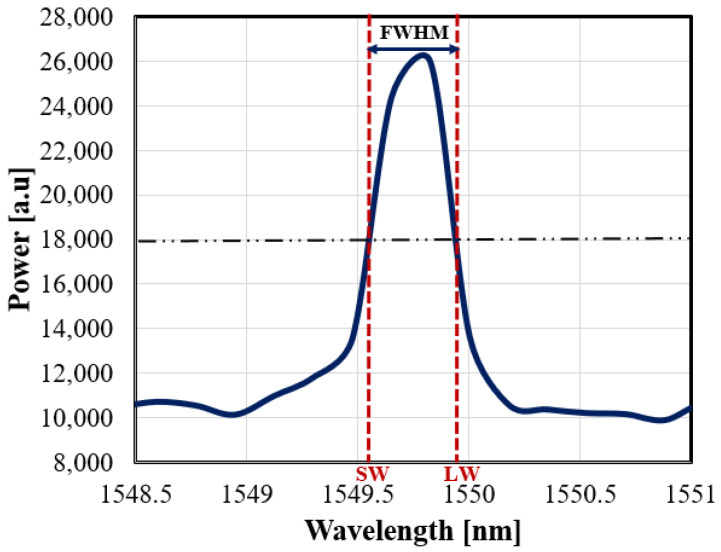
The spectrum of the FBG sensor at room temperature and force-free, the label of the two wavelengths, longer wavelength (LW) and shorter wavelength (SW) are also depicted.

**Figure 3 sensors-21-02002-f003:**
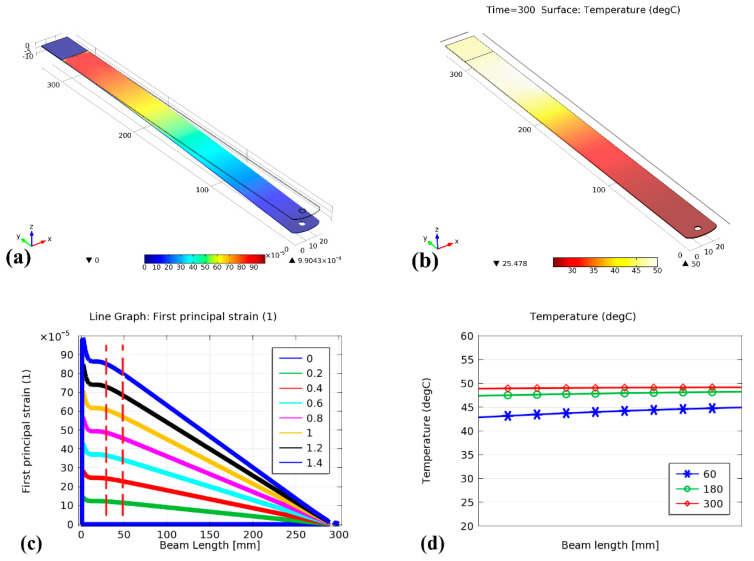
(**a**) Simulation of strain distribution along the beam when 1.4 N is applied, (**b**) temperature distribution along the beam when exposed to the heat source at presumed FBG location, (**c**) strain-induced along the beam length (red lines show FBG edges location), (**d**) temperature distribution at presumed FBG location as time elapsed when the temperature of 50 °C of heat source is exposed.

**Figure 4 sensors-21-02002-f004:**
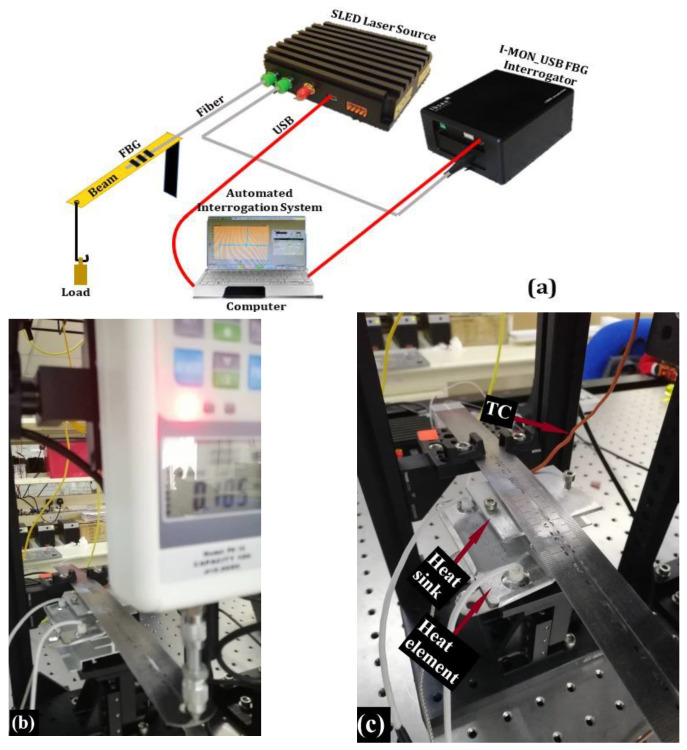
(**a**) Experiment setup schematically illustrating the most critical components of the system, (**b**) the beam with FBG attached subjected to transversal forces and heat through a heat sink, (**c**) illustrates the heat sink to which heat element attached brought closer to the FBG while the system monitors the temperature through thermocouple (TC).

**Figure 5 sensors-21-02002-f005:**
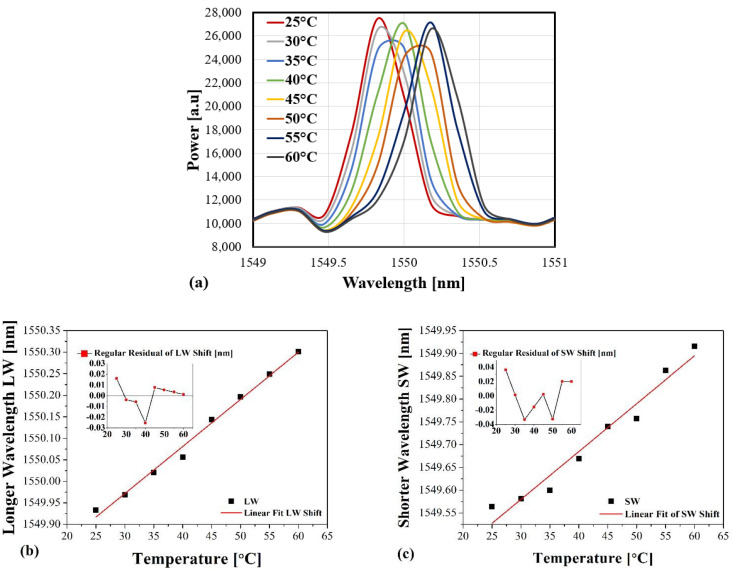
The response of both LW and SW as a function of temperature changes, (**a**) spectrum, (**b**) linear fit of LW and inset illustrating the regular residual shift error, (**c**) linear fit of SW and inset to show the regular residual shift error.

**Figure 6 sensors-21-02002-f006:**
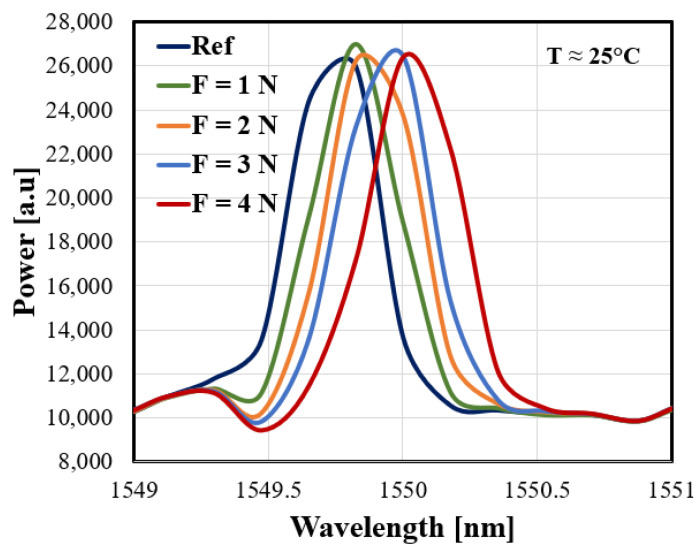
Reflected spectrums of FBG as a function of specific values of transversal forces applied at a constant temperature 25 °C.

**Figure 7 sensors-21-02002-f007:**
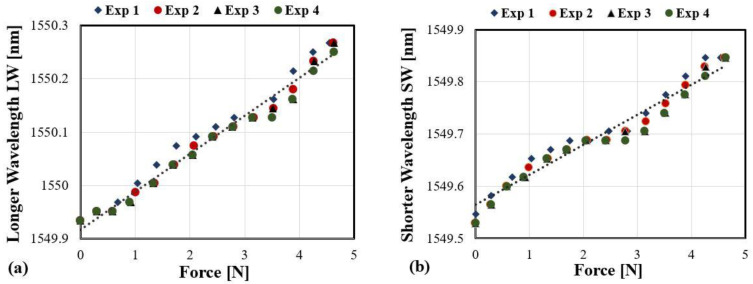
Results of four transversal force calibration procedures, (**a**) the shifts of the longer wavelength vs applied transversal forces, (**b**) the shifts of the shorter wavelength vs applied transversal forces.

**Figure 8 sensors-21-02002-f008:**
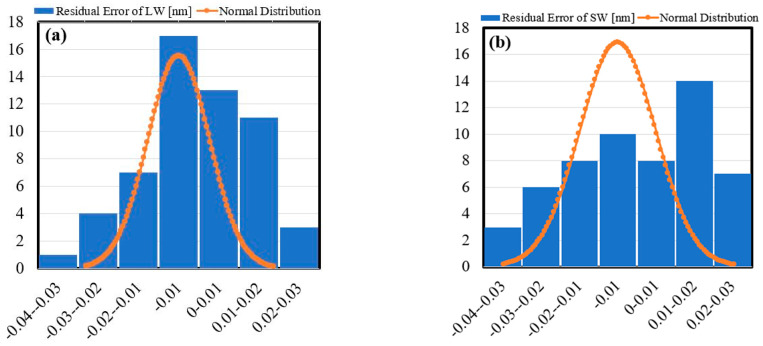
The histogram of the deviated responses from their predicted linear fit points (residual error) of the LW (**a**), and SW (**b**). The superimposed normal distribution function in each shows the variability in our outcome match the distribution or not as a repeatability measurement.

**Figure 9 sensors-21-02002-f009:**
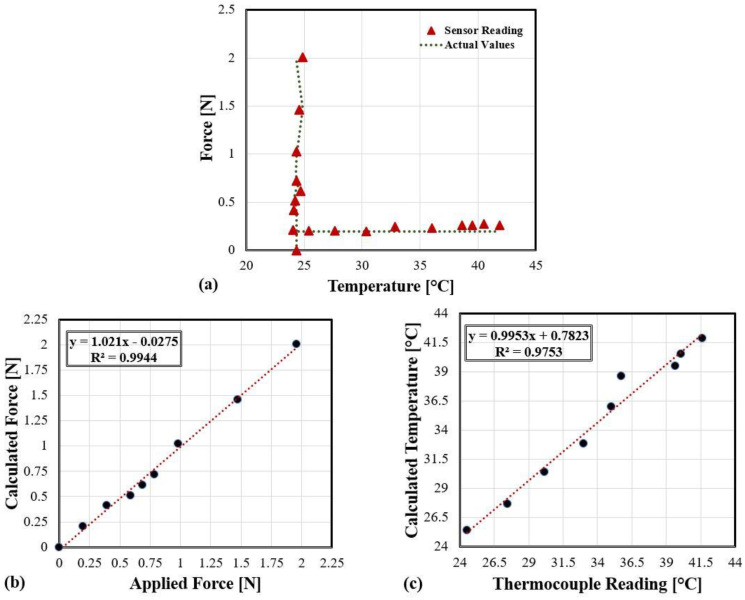
Illustrating the sensor readings compared with the actual values, (**a**) simultaneous measurement of two experiments, (**b**) calculated forces vs applied forces, (**c**) calculated temperature vs. thermocouple readings.

**Table 1 sensors-21-02002-t001:** Physical dimensions and properties of the beam.

Properties/Dimensions	
Length [cm]	35
Width [cm]	2.6
Thickness [cm]	0.07
Young’s Modulus [Gpa]	200
Distance from neutral to the surface (R) [cm]	0.035
Calculated area moment of inertia (I) [m^4^]	7.43 × 10^−13^
Poisson’s ratio	0.3

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
