# Peer review of "Cantilever Beam with a Single Fiber Bragg Grating to Measure Temperature and Transversal Force Simultaneously"

_sensors, 2021, doi:10.3390/s21062002_

Round 1
Reviewer 1 Report
In this article, the authors introduce a new interrogation method that can discriminate temperature and applied transversal forces by analyzing the shifts of the longer and shorter wavelength of a single FBG. This research use LW, SW, and center wavelength rather than using the BW and center wavelength. The researchers use finite element analysis(FEA) in simulation to investigate the force-stain relationship, stain distribution along the beam, and the rate of heat conduction so that they can decide the position of the FBG based on the simulation results. Several suggestions are supplied:
1. Figure 3(a)&(b): Some numbers overlap slightly.
2. Figure 5: The spacing between the three pictures is too small and the typesetting is in a little mess.
3. Figure 6: Suggest mark the exact value of the constant temperature.
4. Suggest the authors improve the discussion part.
Suggest the authors discuss more the drawbacks of the gluing process mentioned in line 236, or how to solve the problem mentioned in line 294.
5. Suggest the authors add more references on this topic, several references are recommended :
Latest Achievements in Polymer Optical Fiber Gratings: Fabrication and Applications MDPI Photonics
Reviewer 2 Report
The authors study a method for measuring temperature and transversal forces in a cantilever beam by using an FBG. I understand that the results may be of interest to the sensing community. In the following, I provide a set of questions/suggestions which may help the authors to improve the quality of their manuscript.
- Would the use of an LPG bring any improvement for the explored method (since LPG spectral signature is typically broader and, perhaps, the spectral asymmetries would be more evident; also, LPGs are typically longer than FBGs)? Please comment on it.
- The wavelength shift data in Fig. 7 shows an oscillatory behavior around the linear trend. Could the authors comment on it?
- In Eq. 8, one sees that the strain coefficients are similar. Please include the error of such coefficients, as estimated from the experimental data fit, for example. It will allow the reader to check if they are distinguishable considering the experimental errors. This is important to validate the proposed procedure.
- In the section “validation experiments”, the authors demonstrate the operation of their procedure for constant temperature and constant applied force situations. However, if the authors could demonstrate a situation in which both force and temperature vary (even if it were for a few data points), it would add much value to their manuscript.
- English language use can be generally improved.
- The picture in Fig. 4b is overlapping with the description in Fig. 4a. Please fix it.
Reviewer 3 Report
Authors presented a FBG-based system for transverse force and temperature sensing, where they analyze the shorter and longer wavelength, which characterize the FWHM analysis. The paper needs major improvements before being considered for publication, the key issues are listed below.
- Major improvements on English usage is mandatory. The paper is very confusing and some sentences are very hard to follow, a native speaker or a professional editing service should be considered.
- Authors should clarify the drawbacks of the proposed technique, since it seems to be only applicable in beam-like structures with one end fixed. The method seems to be not applicable in structures with uniform stress/strain distribution.
- Similarly, it seems the variation of stress/strain along the structure must occur within the FBG physical dimensions, it should be discussed in the paper.
- Actually, this approach uses a linear (or quasi-linear) variation on the strain in the grating region that result in something like a "chirp effect" on the FBG, which broadens its spectrum. As can be observed in Figure 2, the use of shorter and longer wavelengths is similar to use FWHM and center wavelength. The use of FWHM in conjunction with center wavelength for multiparameter monitoring is commonly used in chirped FBGs for different applications see [A]. This should be discussed in the introduction, which lacks on many compensation techniques and other techniques for simultaneous assessment of different parameters.
- Even with uniform FBGs inscribed in standard SMFs, the use of spectral features for temperature and force monitoring was already proposed (see [B] ). Thus, the authors cannot claim that it is the first time that such technique is used. The fact of using two wavelengths (SW and LW) instead of the FWHM and center wavelength does not make this approach completely novel.
- In the same scope as my last comment, the use of spectral features of uniform FBGs (inscribed in polymer optical fibers) was already proposed for simultaneous assessment of multiple loading conditions (strain, torsion and bending) see [C].
- Authors should also consider improving the validation tests, the results shown in Figure 9(a) indicates that both parameters (force and temperature) were not simultaneously varied, since when the force variation occurs, the temperature is constant and vice-versa.
[A] Vorathin, E., Hafizi, Z.M., Aizzuddin, A.M., Zaini, M.K.A., Lim, K.S., 2019. A Novel Temperature-Insensitive Hydrostatic Liquid-Level Sensor Using Chirped FBG. IEEE Sens. J. 19, 157–162. https://doi.org/10.1109/JSEN.2018.2875532
[B] Vorathin, E., Hafizi, Z.M., Aizzuddin, A.M., Lim, K.S., 2018. A natural rubber diaphragm based transducer for simultaneous pressure and temperature measurement by using a single FBG. Opt. Fiber Technol. 45, 8–13. https://doi.org/10.1016/j.yofte.2018.05.011
[C] Leal-Junior, A.G., Theodosiou, A., Diaz, C.R., Marques, C., Pontes, M.J., Kalli, K., Frizera, A., 2019. Simultaneous measurement of axial strain, bending and torsion with a single fiber bragg grating in CYTOP fiber. J. Light. Technol. 37. https://doi.org/10.1109/JLT.2018.2884538
Round 2
Reviewer 1 Report
Suggest accept it
Reviewer 2 Report
The authors have adequately answered my questions.
Reviewer 3 Report
Authors addressed all my comments. I recommend the publication of this work.